# The decentralization effects of entrepreneurial characteristics on corporate social responsibility

**Youqiang Ding**, **Yufeng Hu** *

School of Finance, Tongling University, Tongling, Anhui, China

* huyufenghzu@163.com

## Abstract

The question of whether appropriate decentralization can solve Corporate Social Responsibility (*CSR*) misplacement caused by Entrepreneurial Characteristics (*EC*) is an interesting ethical puzzle. Because corporate behavior depends on the decision-making choices of executives whose personality characteristics affect the choice tendency, power distribution undoubtedly becomes a big boost for most businesses to work out the adverse externality problems. Based on Upper Echelons Theory, this study developed a comparative impact model linking the effects of entrepreneurial intrinsic nature and experience characteristics on *CSR* performance. We tested the effective mechanism with the mediator role of the Corporate Power Distribution Index (*CPDI*) through a sample of listed Chinese companies from 2009 to 2017. The results provide that *EC*, such as female *Gender*, *Degree*, and *Salary*, have positive effects on *CSR*; *CPDI* plays a mediator role in the relationship between *EC* and *CSR*; and is moderated by *Age*, *Academy*, and *Shares*. The conclusion shows that *EC* can improve *CSR* performance to optimize *CPDI* to reduce corporate misplacement behavior.

## Introduction

The unrestrained centralization makes enterprisers dogmatically into principal-agent problems of power-expanded behavior. But over-decentralization also creates buck-passing problems in *CSR* activities. Although the sustainable development of *ESG* (Environment, Social, and Governance) has highlighted *CSR* performance, the result was still unsatisfactory for listed companies' false information disclosure and financial fraud behavior. So, what makes entrepreneurs unable to perform their responsibilities effectively? From the perspective of corporate governance, we try to find a better way to solve this problem by optimizing *CPDI* to balance the infinite expansion of power boundary and the negative impact of *EC* changes.

Simon [1] believed that the decision-making of enterprises is affected by executive ability with incomplete information and proposed a bounded rational hypothesis that provides a basic theory of decision-making based on preliminary data for management studies. This theory is entirely different from the sensible economic assumption. Later, Lieberson and Oconnor

**Funding:** Funding: This research was funded by the Major Project of Higher Educational Humanity and Social Sciences Foundation of Anhui Province (SK2021ZD0084), and the Scientific Research Foundation for Talent Introduction of Tongling University (2021tlxyrc06) for the support during this research. The funders had no role in study design, data collection and analysis, decision to publish, or preparation of the manuscript.

**Competing interests:** Competing interests: The authors have declared that no competing interests exist.

[2] found that CEO leadership change would not affect corporate strategic decision-making. Previous studies seemed to overestimate the decisive role of the entrepreneur's ability. Relying on the principal-agent theory, companies entrust the production, operation, and management authority to professional managers. They confirmed managers' value creation and paid them compensation according to performance evaluation [3, 4]. But few companies had gone out of business even when the CEO's position changed frequently. Therefore, some scholars thought there might be no direct correlation between *EC* and corporate performance over a long period [5].

It was not until 1984 that Hambrick and Mason [6] proposed an Upper Echelons Theory to criticize the mainstream thought of management study, firstly putting *EC* factors into the corporate strategic performance. Then, in 2007, Hambrick [7] extended the theory that the corporate cognition of environmental factors was determined by *EC*, influencing the formulation, selection, and implementation of corporate strategy, which will indirectly affect the achievement of corporate performance. Since then, much more research has described a controversial discussion on the impact mechanism of corporate performance from *EC*, such as age, gender, education, work, motivation, etc., resulting in diverse impacts and different conclusions [8, 9]. In a word, theorists believe that the personal values of executives affect their cognition and preference, which makes *EC* form a unique leadership style and decision-making logic for corporate strategic behavior and then determine enterprises' structure, conduct, and performance [10, 11].

As the Upper Echelons Theory expands the organizational behavior from perfect rationality to bounded rational decision-making, previous studies focused on the relationship between *EC* and *CSR* but ignored the entrepreneur's decision conduct and less considered the mediating effect of *CPDI*. And meanwhile, the effective implementation of *CSR* is also subject to the corporate system of decentralization and authorization. So, this paper studied the influence of *EC* on *CSR* performance from Intrinsic Nature, Experience Characteristics, and Incentive Features. Then we analyzed the mediating effect of *CPDI* on the relationship and discussed the practical path by optimizing them to improve *CSR* performance, which has some enlightenment for corporate strategic research.

## Theory and hypothesis

### The definition of corporate social responsibility

According to the definition by Oliver Sheldon, *CSR* is an enterprise's responsibility to meet society's needs, including legal and moral factors [12]. However, it was not until the 1970s that *CSR* research gradually formed a theoretical system. Since then, *CSR* has generally experienced three evolution stages: Concentric Axis, Pyramid, and Triple Bottom Line models. First, scholars extended *CSR* meaning to corporate conduct. They perform social contracts, balance the relations between society and enterprises, and pursue value maximization [13]. Then, *CSR* is mainly reflected in two aspects: one is not doing evil, which requires entrepreneurs to abide by the law; the other is doing good things, which requires them to correct their conduct by morality.

Martínez et al. [14] revealed that *CSR* behavior is a strategy for the legitimization and survival of companies through institutional and stakeholder perspectives, which needs exploring externally and internally. The research of Xu et al. [15] also believed that *CSR* is generally affected by altruism, motivation, and legitimacy. So, this paper divided *CSR* into three dimensions: (i) entrepreneurs make incomplete rational decisions to undertake *CSR* from emotional considerations, although they may not have an obligation to bear additional social responsibility without compensation; (ii) *CSR* is the pursuit of self-worth because the incentive feature

and experience characteristics of entrepreneurs will stimulate self-actualization needs based on the theory of Maslow's hierarchy of needs; (iii) most entrepreneurs are consciously doing things just as the moral behavior can brings them more benefits results. Excellent *CSR* activities can improve employee performance and financial performance, as well as "moral window-dressing" and "self-interest tools" [16, 17].

## The influence of entrepreneurial characteristics

In terms of emotion, morality, and value, we divided *EC* into three categories: Intrinsic Nature (including *Gender* and *Age*), Experience Characteristics (including *Politics*, *Academy*, and *Degree*), and Incentive Features (including *Salary* and *Shares*).

**(i) Intrinsic nature.** *Gender*. The male executive has strategic thinking to innovate, but the female executive is not good at innovation because of lacking risk spirit [18]. The gender identity theory summarized that female characteristics as more cautious, conservative, and over supervision, while the gender role theory believed that female executives seem to have male characteristics [19]. Du et al. [20] found that enterprises with female CEOs are more conducive to engaging in *CSR* activities. Xu et al. [15] also agreed that the executive's gender affects the enterprise's decision-making on *CSR*. Therefore, the increased proportion of female executives can significantly improve *CSR* performance.

*Age*. Entrepreneurs accumulate experience to affect their decision-making behavior with the growth of *Age*. Young executives are not only good at identifying decisions but also at strategic change. Instead, old executives have more social relations resources but lack risk preference and adventurous spirit; their rich experience makes them more mature, calm, and sophisticated. For instance, Zhang et al. [21] research show that *Age* characteristics positively impact *CSR* information disclosure.

**Hypothesis H1.** The female entrepreneur plays a positive role in improving *CSR* performance, which means a positive correlation between intrinsic nature and *CSR*.

**(ii) Experience characteristic.** The experience characteristics of entrepreneurs include growing, learning, profession, etc. Especially, academic experience makes decision-making more professional and cautious. There is a significant positive correlation between *Political* background and *CSR* performance. The diversity of management can also improve *CSR* performance [22].

*Degree*. High-educated executives have better innovative attitudes and more vital personal ability as a constituent element of human capital. They can quickly capture information from excellent managers to correctly predict decision-making [6]. Cacioppo et al. [23] found that executives with good education and vital profession will increase *CSR* performance. Much research has proved that high-educated entrepreneurs will pay more attention to *CSR* information disclosure, making more rational and objective decisions. Zhang et al. [24] pointed out that the heterogeneity of executive education positively correlated with *CSR* performance.

**Hypothesis H2.** High-educated entrepreneurs contribute to *CSR* performance, which positively correlates experience characteristics and *CSR*.

**(iii) Incentive feature.** Generally speaking, excellent professional managers are attracted to join enterprises with high-paying incentives, such as material aspects, equity, trust, tenure, and other incentive factors, which will help the promotion of their *CSR* preference [25, 26]. Barnea and Rubin [27] believed that executive compensation incentives effectively improve *CSR* performance. The research results of Khan et al. [28] provide reliable evidence that

tournament incentives can motivate CEOs to be more socially responsible. Furthermore, Chen et al. [29] pointed out that the CEO's position impacts *CSR*. But there are different views; for example, Liu and Zhu [30] found that the synergy of executive shareholding and internal control systems can promote inefficient *CSR* investment. Zahid et al. [31] also found that CEO turnover and duality negatively moderate the relationship between corporate financial performance and *CSR* performance. Therefore, we should discuss the effectiveness of incentive features on *CSR* performance.

**Hypothesis H3.** The high *Salary* of entrepreneurs contributes to *CSR* performance, which makes a positive correlation between the incentive feature and *CSR*.

### Mediator role of corporate power distribution index

The power-expanded behavior of entrepreneurs leads to corporate distortion of decision-making. The study of Banerjee et al. [32] shows that the unclear power boundary and ultra vires behavior play a vital role in adverse *CSR* events. Unchecked power expansion creates false accounts, inflated profits, and misleading disclosure. At present corporate governance, there is an overconfidence power of shareholders with an omission of directors and supervisors, who can not play their proper roles but negatively affect accounting conservation [33]. Therefore, it is more likely to breed self-interested behavior of harmful decisions, although beneficial to the centralization of leaders. As a result, entrepreneurs selectively implement their *CSR* activities depending on corporate performance and regulation intensity [34, 35].

*CPDI* can adjust the power boundary. It is crucial to ensure effective decision-making, supervision, and incentive mechanisms [36]. Khan et al. [37] found that the environmental performance of Chinese enterprises is affected by board size, independence, gender diversity, and CEO duality, which positively moderate the relationship between corporate governance and *CSR*. The critical factor is designing the governance structure and operation system to compartmentalize authority. So, there is a separate power between the chairman and general manager to a clear boundary among decisions, management, and supervision in corporate governance. According to Triana et al. [38] and Attig et al. [39], their separate power does not affect corporate performance. At the same time, the functional conflict between directors and supervisors must strictly redefine the power boundary. The present research shows that a reasonable governance structure can reduce the information asymmetry problems in the principal-agent system [40]; an appropriate shareholding ratio can curb the negative consequences of executives' self-interested behavior. And then excellent corporate governance also positively modifies *CSR* performance [41].

Relevant research shows that executive tenure, compensation, ownership, and gender characteristics significantly improve *CSR* performance and reduce the information asymmetry problems of the principal-agent relation [42]. That is to say, high and robust incentives of *CPDI* can increase *CSR* performance. Meanwhile, executives' gender and long-term salary also positively affect environmental performance to avoid regulation sanctions [43, 44]. In addition, executives who prefer social performance goals will take action to reduce ecological pollution [45]. *CPDI* balances the power and willfulness of entrepreneurs to play an influential role in *CSR* performance.

**Hypothesis H4.** *CPDI* can mediate the relation between *EC* and *CSR*.

This study analyzed the impact of Intrinsic Nature and Experience Characteristics with the moderator of Incentive Features and the mediator of *CPDI* (Fig 1).

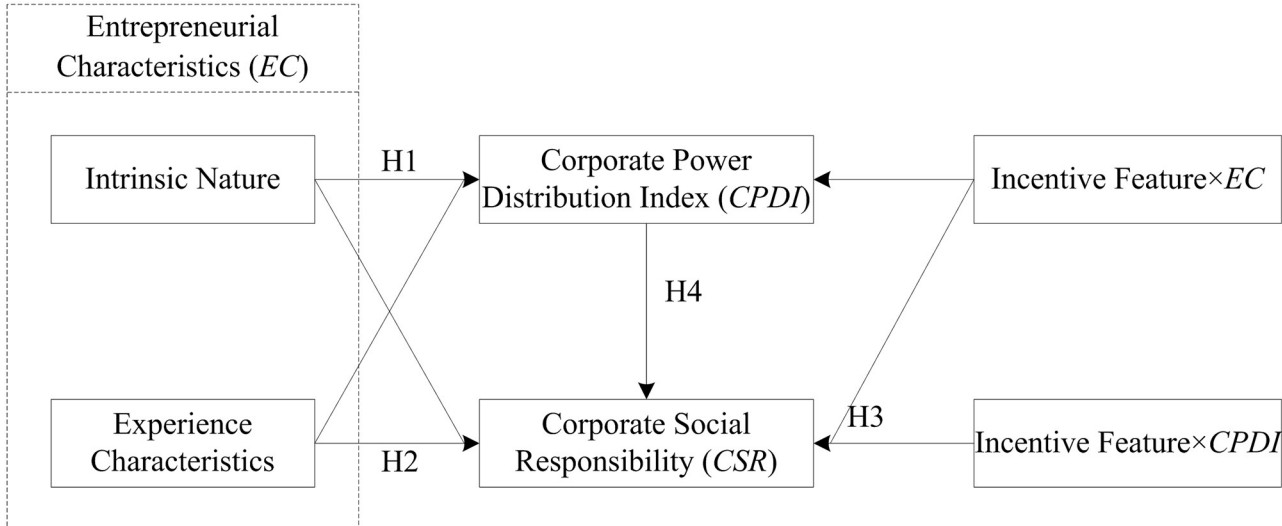

**Fig 1. The mediating effect of *CPDI* on the relation between *EC* and *CSR*.**

## Method

### Empirical model

The test model constructed given the above theoretical analysis is as follows:

$$
\begin{aligned}
CSR \quad &= \beta 0 + \beta 1\, EC + \beta 2\, CPDI + \beta 3\, IF + \beta 4\, EC \times CPDI \\
&\quad + \beta 5\, EC \times IF + \beta 6\, CPDI \times IF + \beta 7\, EC \times CPDI \times IF \\
&\quad + Controls + f + \varepsilon
\end{aligned}
\tag{1}
$$

Where *CSR* is the dependent variable for Corporate Social Responsibility. *EC* is the independent variable of Entrepreneurial Characteristics, including Intrinsic Nature represented by *Gender* and *Age* and Experience Characteristics described by *Degree*, *Academy*, and *Politics*. *IF* stands for Incentive Feature as moderator variables defined by *Salary* and *Shares*. A mediating variable of *CPDI* represents the Corporate Power Distribution Index. *Controls* represents a set of control variables. $\varepsilon$ represents a random interference term.

### Sample selection

Firstly, we obtained *CSR* data of China A-share companies in the ranking reports by Rankins *CSR* Ratings (RKS) from 2009 to 2017. Then, the CEO samples were described by *EC* data with the complement of *Salary*, *Shares*, and *Degree* mainly from CSMAR and RESSET databases. Finally, this study eliminated some delisting shares and undetermined information and selected 2061 valid samples to test Hypothesises.

### Variables measurement

Following the prior literature [46], we use the entropy index method to calculate the seats of directors for *CPDI* as follows,

$$
CDPI = \sum S_i \times \ln\left(\frac{1}{S_i}\right)
\tag{2}
$$

Where *Si* represents the seat proportion of the company, then we used the positions to calculate the entropy value representing *CPDI*. Theoretically, *CDPI* has two parts: The one hand is the division of power in teams, called team power factor (*TPF*); the other is the division of power in individuals, called personal power factor (*PPF*). Therefore, use the following three steps to calculate *CPDI*:

First of all, we divided all executives into three groups: director, supervisor, and management team to calculate *TPF*; then, using 1,···, n-1 to represent the seat number in an institution as the leader percentage of each group to calculate *PPF*; finally, *CPDI* is equal to *TPF×PPF*. Based on previous research, the smaller entropy value represents the more dispersed power of decentralization and authorization for executors.

As a result of the above theoretical research with the influencing factors of *EC* on *CPDI*, we take the control variables as the following factors: the Enterprise's Total Assets (that are owned or controlled for economic benefits), Employee Size (refers to the total number of enterprise jobs, including operators, managers, and employees), Current Assets (that can liquidate within a year or a business cycle), Capital Reserve (refers to the provident fund formed by the acceptance of donations, the premium of equity and the appreciation of property), Operating Cost (refers to the cost of goods or services), and Asset-liability Ratio (reflecting the proportion of assets provided by the enterprise creditors). The data of these variables are obtained from the financial statements of listed companies to take as logarithms.

## Descriptive statistics

Table 1 presents the descriptive statistics of core variables. For the average in our sample, *CSR* equals 38, and *CPDI* equals 0.22. It shows that the contribution of listed companies to *CSR* is quite different. Some research has shown that better companies contribute more to *CSR* performance [26].

CEO's *Age* is equal to 50, and *Degree* is similar to 18, which is more biased towards mature and stable characteristics in an energetic and experienced period. The *Gender* proportion of female CEOs in listed companies is less than 5%. In addition, CEOs' *Salary* and *Shares* are uncertain with in-disclosure information, resulting in a null value.

## Results analysis

### Influence test of entrepreneurial characteristics

Panel A of Table 2 shows that the coefficient of *Gender* is positive and significant at 0.01, which is consistent with the conclusion that the increased proportion of female executives

**Table 1. Descriptive statistics of core variables.**

| Categories | Variables | Mean | Std. | Min | Max |
|---|---|---|---|---|---|
| Dependent Variable | *CSR* | 38.046 | 13.400 | 15.200 | 87.948 |
| Mediator Variable | *CPDI* | 0.220 | 0.028 | 0.134 | 0.316 |
| Independent Variables | *Age* | 49.672 | 5.618 | 34.000 | 68.000 |
| | *Gender* | 0.044 | 0.206 | 0.000 | 1.000 |
| | *Salary* | 13.391 | 0.900 | 8.566 | 16.639 |
| Moderator Variables | *Shares* | 12.626 | 2.928 | 4.606 | 20.298 |
| | *Degree* | 17.873 | 1.504 | 12.000 | 19.000 |
| | *Academy* | 0.107 | 0.310 | 0.000 | 1.000 |
| | *Politics* | 0.140 | 0.347 | 0.000 | 1.000 |

*Gender* is set to 1 if CEO is female; otherwise, 0. *Age* is equal to the CEO's age. Observation: 2061.

**Table 2. The effects of entrepreneurial characteristics on corporate social responsibility.**

| Variables | Intrinsic Nature (A) | | | Experience Characteristics (B) | | | Incentive Feature (C) | | | (D) |
|---|---|---|---|---|---|---|---|---|---|---|
| | (1)CSR | (2)CPDI | (3)CSR | (4)CSR | (5)CPDI | (6)CSR | (7)CSR | (8)CPDI | (9)CSR | (10)CSR |
| Gender | 0.058# (1.91) | 0.017*** (6.56) | 0.081*** (2.72) | | | | | | | 0.080*** (2.71) |
| Degree (16) | | | | 0.125* (2.06) | -0.008*** (−3.13) | 0.116# (1.84) | | | | 0.102# (1.74) |
| Degree (19) | | | | 0.157*** (2.60) | -0.009*** (−3.60) | 0.147* (2.34) | | | | 0.135* (2.31) |
| Salary | | | | | | | 0.056*** (8.01) | -0.002*** (−3.14) | 0.054*** (7.79) | 0.054*** (2.31) |
| CPDI | | | -1.345*** (−5.74) | | | -1.233*** (−5.33) | | | -1.008*** (−4.36) | -1.072*** (−4.54) |
| Controls | Y | Y | Y | Y | Y | Y | Y | Y | Y | Y |
| adj.$R^2$ | 0.336 | 0.204 | 0.347 | 0.338 | 0.188 | 0.347 | 0.353 | 0.190 | 0.359 | 0.364 |

# significance at 0.10,

* significance at 0.05,

** significance at 0.01,

*** significance at 0.001. The *t* value is enclosed in parentheses. *Controls* include control variables, year-fixed effect, and firm-fixed effect.

promotes *CSR* performance in literature. As the proportion of female executives in listed companies is relatively small, they pay more attention to their social identity and personal accomplishment through *CSR* implementation. The mediator impact of *CPDI* has a significant positive correlation with female *Gender* characteristics ($\beta$=0.017, $p$<0.05, Model 2). It indicated that female executives prefer stable structures or are not good at changing corporate governance. And it has a negative effect on *CSR* and is significant at 0.01 (Model 3). This support the Intrinsic Nature of CEOs towards *CSR* for Hypothesis H1.

Panel B of Table 2 shows that the *Degree* coefficient has a positive impact on *CSR*, which depends on the fact that a graduate degree ($\beta$=0.157, $p$<0.001, Model 4) has a more significant effect than a college degree ($\beta$=0.125, $p$<0.05). The CEO's learning by education or by doing enhances the ability to distinguish, judge, and recognition, making correct decisions for the consideration of corporate interests. This support the Experience Characteristics of CEOs towards *CSR* for Hypothesis H2.

Panel C of Table 2 shows that the characteristics of *Salary* have a positive effect on *CSR* ($\beta$=0.056, $p$<0.001, Model 7). The higher the CEO's salary, the stronger the material satisfaction. On the one hand, they have the financial strength to do good deeds; on the other hand, they do good deeds to meet high-level needs, and corporate social responsibility is relatively easy to fulfill. This support the Incentive Feature of CEOs towards *CSR* for Hypothesis H3.

For the impact of *CPDI*, female *Gender* characteristics have a significantly positive effect on *CPDI* ($\beta$=0.017, $p$<0.001, Model 2), indicating that female executives prefer a stable corporate governance structure. In other words, they are not good at innovation or change in corporate governance. There is a negative correlation between *Degree* characteristics and *CDPI* ($\beta$= −0.008 and $\beta$=−0.009, $p$<0.001, Model 5). They will be more inclined to optimize the governance structure, which comes from professional self-confidence and management ability to enhance the highly educated entrepreneurs who are better at delegating. There was a negative correlation between the *Salary* characteristics and *CPDI* ($\beta$=−0.002, $p$<0.001, Model 8), according to the distribution of work to undertake more tasks and responsibilities. The larger enterprise would need more senior professional managers. In addition, a negative correlation

exists between *CPDI* and *CSR* ($\beta=-1.072$, $p<0.001$, Model 10), which shows that improving *CPDI* can promote *CSR* quality. In other models, the significance result remains unchanged. Therefore, these results support hypotheses H1, H2, and H3.

## Mediator test of corporate power distribution index

We now test Hypothesis 3 and 4, which concern boundary conditions of incentive policy and power restrictions for CEOs. To investigate the moderating effect of the Incentive Feature, we focus on the *Salary* coefficient and its interaction with *Shares*. Table 3 shows the empirical results. It shows that the coefficients of *Salary* and *Gender* are positive and significant at 1%, suggesting that the implementation of compensation incentives for female executives seems to play a positive role.

The variable of interest as *Salary×Shares* is positive and significant at 10% (Model 2), demonstrating that both *Salary* and *Shares* are given to make *CSR* performance regardless of *Gender*. Presumably, the paid CEOs with equity will consider the benefits of shareholders. However, the interaction of *Gender×Salary* (Model 3) is negative and insignificant, demonstrating no significant difference between male and female CEOs in terms of their preference for salary incentives. On the other hand, when we give a *Shares* incentive, the interaction of *Gender×Shares* (Model 1) is negative and significant at 5%, indicating that female executives' equity ownership can restrain *CSR* performance.

**Table 3. The moderating effect of the corporate power distribution index.**

| Variables | *Shares* × CPDI (**A**) | | | | *Age* × CPDI (**B**) | | | | *Academy* × CPDI (**C**) | | | |
|---|---|---|---|---|---|---|---|---|---|---|---|---|
| | (1)*CSR* | (2)*CPDI* | (3)*CSR* | (4)*CSR* | (5)*CSR* | (6)*CPDI* | (7)*CSR* | (8)*CSR* | (9)*CSR* | (10)*CPDI* | (11)*CSR* | (12)*CSR* |
| *Salary* | 0.036*** (3.96) | -0.002* (−2.30) | 0.033*** (3.68) | 0.035*** (3.86) | | | | | | | | |
| *Shares* | 0.009*** (3.43) | 0.002*** (6.32) | 0.011*** (4.19) | -0.024*** (−1.19) | | | | | | | | |
| *Gender* | | | | | 0.058# (1.95) | 0.018*** (6.58) | 0.084*** (2.79) | 0.083*** (2.79) | | | | |
| *Age* | | | | | 0.165*** (3.11) | 0.023*** (4.63) | 0.198*** (3.76) | 0.883*2.43) | | | | |
| *Degree*(16) | | | | | | | | | 0.121*2.00) | -0.008*** (−3.20) | 0.112# (1.78) | 0.109# (1.72) |
| *Degree*(19) | | | | | | | | | 0.151*2.49) | -0.009*** (−3.71) | 0.140*2.23) | 0.134*2.14) |
| *Academy* | | | | | | | | | 0.042* (2.48) | 0.002* (1.13) | 0.045*** (2.59) | -0.350*** (−2.68) |
| *CPDI* | | | -1.358*** (−4.93) | -3.448*** (−2.81) | | | -1.439*** (−6.09) | 10.503# (1.72) | | | -1.246*** (−5.40) | -1.429*** (−5.87) |
| *Shares* × *CPDI* | | | 0.161# (1.79) | | | | | | | | | |
| *Age* × *CPDI* | | | | | | | | −3.061# (−1.96) | | | | |
| *Academy* × *CPDI* | | | | | | | | | | | | 1.793*** (3.10) |
| Controls | Y | Y | Y | Y | Y | Y | Y | Y | Y | Y | Y | Y |
| Observation | 1200 | 1200 | 1200 | 1200 | 2061 | 2061 | 2061 | 2061 | 2061 | 2061 | 2061 | 2061 |
| adj.$R^2$ | 0.341 | 0.175 | 0.353 | 0.353 | 0.339 | 0.212 | 0.351 | 0.352 | 0.339 | 0.188 | 0.348 | 0.350 |

The same as the comments in the previous table.

Table 3 (Model 4) also shows that the coefficient of the variable of interest, *Gender×Salary×-Shares* is positive and significant at 10%, demonstrating that *CSR* has become more prominent in firms with female CEO after the effect of the interaction between *Salary* and *Shares*, which significantly enhances the promotion of *CSR*. This result suggests intense material satisfaction if CEO's salary is high with *Shares* in this sample, which is relatively easy to fulfill *CSR* by doing good things with high-level personal needs, supporting Hypothesis 3.

### Moderator test of incentive feature

We drew Fig 2 to describe further the moderating effects of *Shares*, *Age*, and *Academy*.

i. The main effect between *CPDI* and *CSR* is negative, but the coefficient of *Shares* Incentive Feature is positive (Fig 2a). In addition, the moderating influence of the interactive item between *Shares* and *CPDI* is significantly positive, indicating that the negative effect of *CPDI* on *CSR* becomes weakened by the moderator of *Shares*. The primary cause is that *Shares* can strengthen the CEO's ability to control the company to practice *CSR* activities.

ii. The *Age* coefficient is positive, and the moderating effect of the interactive term between *Age* and *CPDI* is negative, indicating that the negative impact of *CPDI* on *CSR* has increased with the moderating effect of *Age* Intrinsic Nature (Fig 2b). On the one hand, the old CEO's leadership authority comes from decentralized corporate administration, personal prestige, and leadership style. Because their qualification has developed enough in the company, on the other hand, the old CEOs tend to decentralize affairs and authorizations to cultivate new managers.

iii. The influence of the *Academy* on *CSR* is not significant (Fig 2c). Still, the moderating effect of the interactive term between *Academy* and *CPDI* is significantly positive, indicating that the negative impact of *CPDI* on *CSR* becomes weakened with the moderating effect of *Academy* background characteristics.

Fig 2 also shows a significant positive correlation between *Politics* characteristics of entrepreneurs and *CSR*. We found that the moderating effect of the interactive term between *Politics* and *Academy* backgrounds is significantly negative, indicating that the moderator of *Politics* background characteristics weakened the positive impact of the *Academy* on *CSR* performance. *Politics* or *Academy* knowledge of the CEO will make them more accurately recognize and grasp the boundaries of responsibility between the company and government from a

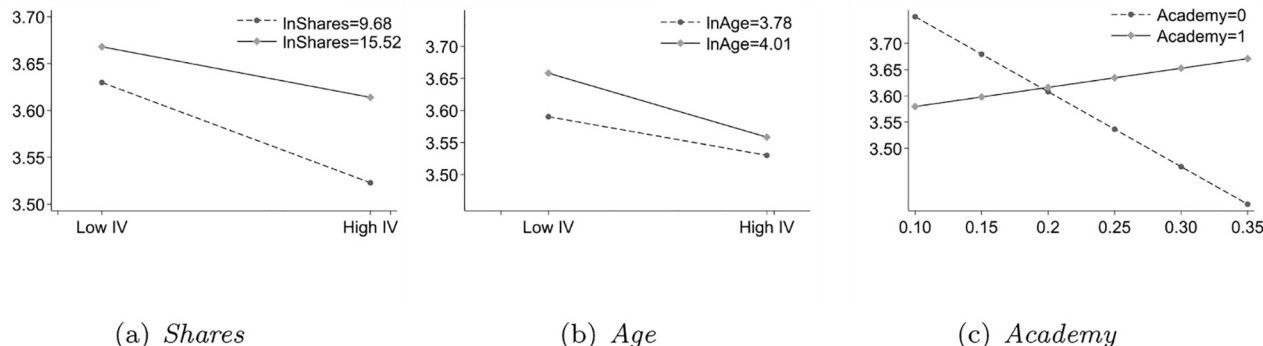

(a) *Shares*          (b) *Age*          (c) *Academy*

**Fig 2. The moderating effects of incentive feature.** The vertical axis shows *CSR*, and the horizontal axis shows *CPDI*. (a) Shares, (b) Age, and (c) Academy.

professional perspective. The government wants to control *CSR*, but the company is only pursuing business performance.

Then, we can see that *CPDI* is a moderated mediator variable, which supports Hypothesis 4.

## Discussion

Although we have considered the fixed effects of years and firms in the above inspection process, many endogenous problems still exist due to the heterogeneity of companies. The main reason is that high-salary companies tend to be more excellent for *CSR* performance. A good *CSR* company also attracts more excellent managers requiring a high salary. In addition, the moderator of *CPDI* cannot be directly observed and calculated by the entropy index method and may have missing variables or measurement errors. So the estimated relationship is not accurate enough to result in endogenous problems. Therefore, we use the instrumental variable method to estimate the endogenous issues of *Salary* and *CPDI*.

Firstly, we used the instrumental variables with lag 1 *CSR* to report in Table 4. The empirical results show that *Salary* characteristics and *CPDI* have passed the test. The results of phase I of the estimation results (Model 1 and Model 2) show a significant positive correlation between lag 1 *CSR* and *Salary* ($\beta$=0.527, $p$ <0.001) and a significantly negative correlation with *CPDI* ($\beta$ =−0.012, $p$<0.001), which is consistent with the previous test results. In Phase II of the estimation results (Model 3 and Model 4), *Cragg-Donald Wald F* is 28.759 and 51.922, and the $p$-value of *KP rk LM* is less than 0.001. Therefore, this result indicated no weak instrumental variables and unrecognizable problems in the models. So, it has passed the first phase requirements.

**Table 4. Regression results for instrumental variables.**

| Variables | Examination of Phase (I) | | | | | | Examination of Phase (II) | | | |
|---|---|---|---|---|---|---|---|---|---|---|
| | (1)*CSR* | (2)*CPDI* | (3)*Salary* | (4)*CPDI* | (5)*Salary* | (6)*CPDI* | (7)*CSR* | (8)*CSR* | (9)*CSR* | (10)*CSR* |
| *lag*1.CSR | 0.527*** (6.97) | -0.012*** (−5.51) | | | | | | | | |
| *Distance* | | | -0.106*** (−10.85) | -0.001*** (−4.02) | -0.092*** (−9.94) | -0.001*** (−3.86) | | | | |
| *EQ* | | | -0.282*** (−5.56) | 0.008*** (4.75) | -0.231*** (−4.62) | 0.009*5.73) | | | | |
| *MS* | | | 0.366*** (7.12) | -0.006*** (−4.14) | 0.273*** (5.55) | -0.008* −5.43) | | | | |
| *Salary* | | | | -0.002*** (−3.54) | | -0.003*** (−3.76) | | 1.551*** (6.99) | 0.282*** (8.53) | 0.300*** (7.53) |
| *CPDI* | | | -2.634*** (−3.51) | | -2.653*** (−3.72) | | -69.285*** (−5.56) | | -4.484* −2.07) | -3.839* −2.05) |
| *Controls* | Y | Y | Y | Y | Y[①] | Y[①] | Y | Y | Y | Y |
| *Observation* | 2061 | 2061 | 2061 | 2061 | 2061 | 2061 | 2061 | 2061 | 2061 | 2061 |
| *KPrkLM(adj.R* $^2$) | 0.162 | 0.200 | 0.209 | 0.207 | 0.291 | 0.234 | 27.701*** | 46.109*** | 44.336*** | 59.084*** |
| *WaldF(F)* | 47.012 | 57.365 | 52.301 | 47.032 | 49.461 | 34.274 | 28.759 | 51.922 | 13.344 | 17.648 |
| *Stock-Yogo bias* Critical value | | | | | | | 10%:16.38; 15%:8.96 | 10%:16.38; 15%:8.96 | 10%:13.43; 15%:8.18 | 10%:13.43; 15%:8.18 |
| *Hansen J* (Chi-sq) | | | | | | | / | / | 0.150 (0.698) | 0.663 (0.416) |

Models (1)—(6) are first-stage tests with adj.$R^2$ and *F* values are in parentheses. Models (7)—(10) is the second stage test with *Chi-sq* in a square bracket. The control variables in this table are the same as above. In addition, $Y^①$ adds other variables of *EC*.

Secondly, to further test the endogenous problems of the moderating model, we choose the following instrumental variables:

i.   Carbon emission is an instrumental variable used to test environmental quality (*EQ*). Ordinarily, poor air quality reduces brain activity and affects entrepreneurs' physical and mental health, but good air quality makes executives pleasant to create better decisions. The data on carbon emission comes from the IPCC Sectoral Approach.

ii.  Entrepreneur's possession of a healthy body is a prerequisite for its characteristic performance. The fast-paced work makes executives fatigued or even overworked. Subhealth is hindering the entrepreneur's vitality. The executives' health depends on the medical standard (*MS*), presented by the number of hospital beds as a proxy variable. The data comes from the China Health Statistics Yearbook.

iii. Information disclosure distance (*Distance*) between the listed company and the exchange of Shanghai and Shenzhen. And *Distance* is calculated based on the latitude and longitude as the following formula.

$$Distance = \{[(Lng - LngS) \times 85.567]^2 + [(Lat - LatS) \times 111.7]^2\}^{\frac{1}{2}} \tag{3}$$

Where (*Lng, Lat*) is the latitude and longitude coordinates of the listed company, and (*LngS, LatS*) is the latitude and longitude coordinates of the Shanghai and Shenzhen exchanges. Taking the Beijing latitude and longitude coordinates as the benchmark, the one dimension is 111.7 according to the spherical north-south distance $\pi R/180$ km. We can calculate the value according to the spherical east-west distance $\pi R\cos\theta/180$ km. One longitude is equal to 85.567 km.

In Table 4, phase I of the estimation results (Model 3 and Model 4) show that the area with close information disclosure distance, good air quality, and high medical standards positively affects the CEO's salary. In particular, the Yangtze River Delta and Pearl River Delta urban agglomerations have relatively developed economies, medical prototypes, and wages. Due to the different development degrees of the company's management, the possible space for *CPDI* in some eastern coastal areas is more significant than in other regions in China.

Phase II of the estimation results (Model 9) shows a positive correlation between *CSR* and *Salary* ($\beta$=0.282, $p$<0.001) and a negative correlation with *CPDI* ($\beta$ =−4.484, $p$<0.1), which is consistent with the above conclusions. Furthermore, according to the Pule of thumb, *Cragg-Donald Wald F* is 17.648, *KP rk LM* corresponds to $p$ at 0.001, and the Hansen test is insignificant. This result shows no weak instrumental variables or unrecognized and over-identified problems in the models. So, using the instrumental variables to estimate the impact of *Salary* and *CPDI* on *CSR* performance is necessary.

The above empirical results show that:

i.   The *Gender* of Intrinsic Nature has a different impact, where female entrepreneurs contribute to *CSR* performance. But it is mainly due to the low proportion of female executives who try to make a good impression in the public eye. And *Age* is mediating in the relationship between *EC* and *CSR* performance.

ii.  The *Degree* of Experience Characteristics can promote *CSR* performance. The CEO's responsibility sense by educating or doing, especially the impact on graduate students, is more significant than undergraduate students. But entrepreneurs with *Academy* and *Politics* backgrounds have weakened the positive effect of *Degree* on *CSR*.

iii. As the non-responsibility behavior of entrepreneurs mainly comes from self-interested power expansion; we set up two dimensions of encouragement and restriction. From the perspective of the Incentive Feature, *Salary* is still an effective way to stimulate their sense of responsibility and personal value realization in pursuit of self-actualization needs. But it will weaken the responsibility sense of entrepreneurs by giving them stock rights. On the other hand, from the perspective of restriction, *CPDI* has promoted *CSR* performance. That is to say, decentralizing decision-making power benefits the restriction of individual power and self-interest.

## Conclusions and policy implementations

In this study, we used the personal data of CEOs and the financial data of listed companies to research the influence of Entrepreneurial Characteristics (*EC*) on Corporate Social Responsibility (*CSR*) with the moderating role of the Corporate Power Distribution Index (*CPDI*) based on the Upper Echelons Theory. In terms of emotion, morality, and value, we found that the *Academy*, *Salary*, and female *Gender* of *EC* have significant positive effects on *CSR* performance. Under the moderators of *Age*, *Shares*, and *Academy* characteristics, *CPDI* can significantly mediate the relationship between *EC* and *CSR*. Moreover, *Politics* background significantly mediates the impact of a *Degree* on *CSR*. The result provides that corporate strategy should optimize the power distribution of the internal governance structure from entrepreneurs' self-interest to solve the adverse *CSR* events. As entrepreneurs often utilize their capacity and interests to carry out *CSR* activities, both the Incentive Feature and the Experience Characteristics can stimulate their responsibility sense to promote *CSR* performance. So there are two ways to improve *CSR* performance:

On the one hand, it doesn't mean enterprises give up pursuing benefits to fulfill social responsibility activities. In contrast, they ought to optimize the corporate governance structure of the board of directors, supervisors, and managers to improve power distribution to avoid adverse *CSR* events even better for benefits. These checks and balances are affected by entrepreneurial characteristics, which adjust the corporate governance structure from different aspects of improving the educational learning degree of senior executives, introducing directors with academic or political backgrounds, and adequately allocating reasonable gender ratios. In addition, we should implement the incentive system with a management shareholding plan to reduce adverse *CSR* events; only when entrepreneurs know that the value is greater than the loss caused by *CSR* activities will it improve *CSR* performance correctly and efficiently.

On the other hand, the governance of adverse *CSR* events should look into its root cause and symptoms of the problems for the regulatory authorities. In a short time, what plays an influential role is to give entrepreneurs material incentives, such as salary and equity, to encourage them to improve their *CSR* performance more actively and reward them for doing well. At the same time, regulators should note that *CSR* performance reflects more corporate ethics and entrepreneurship, which requires an arduous long-term project at the moral level. Therefore, the government ought to design a set of incentive and restraint mechanisms for entrepreneurs and establish a unique entrepreneurial credit evaluation system by combining morality, interests, and law to promote the enterprises to integrate *CSR* performance into corporate performance.

## Author Contributions

**Conceptualization:** Youqiang Ding.

**Data curation:** Yufeng Hu.

**Formal analysis:** Youqiang Ding.

**Methodology:** Youqiang Ding.

**Validation:** Yufeng Hu.

**Writing – review & editing:** Youqiang Ding.

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
