## [Decision Letter · Decision Letter 0]

20 Jul 2022

PONE-D-22-16972The decentralization effects of entrepreneurial characteristics on corporate social responsibilityPLOS ONE

Dear Dr. Hu,

Thank you for submitting your manuscript to PLOS ONE. After careful consideration, we feel that it has merit but does not fully meet PLOS ONE’s publication criteria as it currently stands. Therefore, we invite you to submit a revised version of the manuscript that addresses the points raised during the review process.

ACADEMIC EDITOR: 

Thank you for your submitted manuscript. As you can see from the report, two reviewers showed opposing views on your paper: one suggested the rejection and one suggested the minor revision. I therefore have carefully considered the manuscript myself to see its potentiality and the possibility of the revision. I personally think that the topic is interesting so I would like to give you another chance to revise and resubmit. I hope this decision will be fair for the authors. If you are happy with the decision, please carefully follow two reviewers' comments and address them as much as you can. I note that this is a major revision decision so the authors should be careful in their revision and respond to each comment in details. The revised manuscript and responses to reviewers may be re-sent to the reviewers for their reconsideration. If the authors fail to satisfy them, the acceptance or minor revision decision for the next round will not be guaranteed. Based on the review reports, I would like to summarise the key following comments for you to easier revise the manuscript. Details should be read in the reports.

**1. The abstract needs an academic hook, that awakes the interest of the readers. **<o:p></o:p>

**2. The motivation of the study needs to be clearer.**

**3. Providing convincing reasons for the controls such as CSR and other variables in the empirical model. **<o:p></o:p>

**4.Consider controlling some corporate governance variables such as the size of the board of directors, auditor quality, board independence, CEO characteristics, etc… (where the data is available)**

**5. Consider controlling for firm/industry fixed effects. Inclusion of time fixed effect is also encouraged.**

**6. Using professional proofreading to improve comprehensive academic writing style and the quality of communication**

**7.Minor errors need to be solved:**

*   In hypotheses 1, there is a mistake after H1 "," (lines 102-103).*

   In conceptual model there is an overlapping of the arrows (Figure 1)

*   Table 1 can start at page 6, instead of the ending of the previous page.*

<o:p></o:p>

We look forward to receiving your revised manuscript.

Kind regards,

Vu Quang Trinh, PhD

Academic Editor

PLOS ONE

Journal Requirements:

2. Please note that PLOS journals require authors to make all data necessary to replicate their study’s findings publicly available without restriction at the time of publication. Please see our Data Availability policy at https://journals.plos.org/plosone/s/data-availability. As such, please make the full specific dataset used in this study available by either A) uploading the full dataset as supplementary information files, or B) including a URL link in your Data Availability Statement and Methods section to where the full specific dataset used in this study can be accessed.

"Funding: This research was funded by the Major Project of Higher Educational Humanity and Social Sciences Foundation of Anhui Province (SK2021ZD0084), and the Scientific Research Foundation for Talent Introduction of Tongling University (2021tlxyrc06) for the support during this research."

5. Thank you for stating the following in the Funding Section of your manuscript: 

"This research was funded by the Major Project of Higher Educational Humanity and Social Sciences Foundation of Anhui Province (SK2021ZD0084), and the Scientific Research Foundation for Talent Introduction of Tongling University (2021tlxyrc06) for the support during this research."

"Funding: This research was funded by the Major Project of Higher Educational Humanity and Social Sciences Foundation of Anhui Province (SK2021ZD0084), and the Scientific Research Foundation for Talent Introduction of Tongling University (2021tlxyrc06) for the support during this research."

7. PLOS requires an ORCID iD for the corresponding author in Editorial Manager on papers submitted after December 6th, 2016. Please ensure that you have an ORCID iD and that it is validated in Editorial Manager. To do this, go to ‘Update my Information’ (in the upper left-hand corner of the main menu), and click on the Fetch/Validate link next to the ORCID field. This will take you to the ORCID site and allow you to create a new iD or authenticate a pre-existing iD in Editorial Manager. Please see the following video for instructions on linking an ORCID iD to your Editorial Manager account: https://www.youtube.com/watch?v=_xcclfuvtxQ

8. Please amend either the abstract on the online submission form (via Edit Submission) or the abstract in the manuscript so that they are identical. 

Additional Editor Comments:

NA

Reviewers' comments:

Reviewer's Responses to Questions

**Comments to the Author**

1. Is the manuscript technically sound, and do the data support the conclusions?

Reviewer #1: No

Reviewer #2: Yes

2. Has the statistical analysis been performed appropriately and rigorously? 

Reviewer #1: No

Reviewer #2: Yes

3. Have the authors made all data underlying the findings in their manuscript fully available?

Reviewer #1: Yes

Reviewer #2: Yes

4. Is the manuscript presented in an intelligible fashion and written in standard English?

Reviewer #1: No

Reviewer #2: Yes

5. Review Comments to the Author

Reviewer #1: Referee report

Manuscript ID: PONE-D-22-16972

The decentralization effects of entrepreneurial characteristics on corporate social responsibility

This study intends to investigate whether appropriate decentralization can solve the misplacement of corporate social responsibility (CSR) caused by entrepreneurial characteristics (EC). The authors find that EC, such as female gender, academic degree, and Salary have positive effects on CSR; CPDI plays a mediator role in the relationship between EC and CSR; and that is moderated by Age, Academic, and Shares.

Although the research question seems interesting, the study suffers from several issues. I list them as the major concerns as follows:

1. Motivation of the study: The motivation of the study is not clear. The authors did not clearly discuss the gap in the literature where their study fits in. Also, there should be some brief discussion about the findings and the contributions of the study in the Introduction as well. This might help position the study and better show why this study matters. I do not find a convincing discussion of the importance of the study in the Introduction section. I suggest the authors to rewrite the whole section.

2. Empirical approach: I do not see convincing reasons for the controls for CSR in the empirical model. The control variables are total enterprise assets, employee size, current assets(?), capital reserve, operating costs, and asset-liability ratio. Why those variables? The authors provided no explanation and no description of how those variables are calculated. In common CSR studies, people would control for some corporate governance variables such as the size of the board of directors, auditor quality, director independence, CEO characteristics, etc… This practice should be applied in this study as well because the study involves decentralization of power in the firm. Moreover, the authors did not control for firm/industry fixed effects (different firms and industries might have different customs, especially in the distribution of power within a firm/ or CSR practices/ or resources for CSR/ or need to CSR-dressing/ or simply because of their business nature and the expose to public media so that they have to build a better image), thus the model might be seriously exposed to the omitted variable bias. Inclusion of time fixed effect is also encouraged.

3. Quality of communication: The manuscript is difficult to read. In my opinion, the manuscript suffers from two problems: lack of professional proofreading, and lack of comprehensive academic writing style.

• After reading the manuscript the first time, I assume that the authors did not get the manuscript proofread. Wrong use of words and verb tenses are very typical in the paper. Frankly speaking, the current quality of writing is not good enough for publication. Please get the manuscript proofread by a professional service.

• About the writing style: I do not find a coherent writing style that can lead readers through the presentation of the study in this manuscript. First, the authors did not put forth a definition of “decentralization”, is it power decentralization? By decentralization, a general reader might come up with any concept. Second, in the Introduction, the authors used three paragraphs to discuss about different issues and the state of literature before mentioning what they are going to do in the paper. I suggest the authors to go straight to what you are going to do in the paper (your research objective/question) as soon as possible in the Introduction. Otherwise, you lose your readers in the 1st page of the Introduction. Third, please do not use past tense in the manuscript, especially you discuss previous studies’ findings. The findings are still there, you should use the past tense if they had already been changed (author revisions or smth like that).

Because of those abovementioned concerns, I am regretted to suggest a Rejection of this manuscript. I hope my comments and suggestions help improving the manuscript. I wish the authors the best with their study.

Reviewer #2: I believe that the paper presents a relevant analysis of an interesting topic related with Corporate Social Responsibility and its effects on organizational benefits. From my point of view there are some minor aspects to solve:

1- The abstract needs an academic hook, that awakes the interest of the readers. Instead of saying that all of these relationships are a puzzle to solve, it will be more interesting to say why this article adds value to the field of research.

I will also add the number of listed companies (sample) in the abstract.

2. Theory and hypotheses. This section is well written and supported in literature. There are some aspects that can easily solved:

In hypotheses 1, there is a mistake after H1 "," (lines 102-103).

In conceptual model there is an overlapping of the arrows (Figure 1)

Table 1 can start at page 6, instead of the ending of the previous page.

All of the figures and tables add value to the research and contribution. For me it's a little bit strange, see the Table 4 in the discussion section, because these last sections (conclusions and discussion ) providence the added value and some insights of the analysis and data presented, but obviously it's a personal decision of the authors.

In general terms it's a research well defended with interesting implications.

6. PLOS authors have the option to publish the peer review history of their article (what does this mean?). If published, this will include your full peer review and any attached files.

Reviewer #1: No

Reviewer #2: **Yes: **Jesus Barrena-Martinez

---

## [Author Response · Author response to Decision Letter 0]

17 Sep 2022

Dear Editors and Reviewers:

Thank you for your letter and for the reviewers’ comments concerning our manuscript entitled “The decentralization effects of entrepreneurial characteristics on corporate social responsibility” (PONE-D-22-16972). Those comments are all valuable and very helpful for revising and improving our paper, as well as the important guiding significance to our research. We have studied the comments carefully and have made a correction which we hope meets with approval. Revised portions are marked in red on the paper. The main corrections in the paper and the responses to the reviewer’s comments are as flowing:

1. The abstract needs an academic hook, that awakes the interest of the readers.

Response: We have rewritten this part of the abstract according to the Reviewer’s comment. The main contributions are as follows: 

“The question of whether appropriate decentralization can solve the misplacement of corporate social responsibility (CSR) caused by entrepreneurial characteristics (EC) is an interesting ethical puzzle. Because corporate behavior depends on the decision-making choices of executives, and their personality characteristics affect the choice tendency. It may be a good way for most businesses to solve the negative problems of CSR. ”

2. The motivation of the study needs to be clearer.

Response: Based on the negative events of corporate social responsibility (CSR), this paper studies the influence of corporate executives’ characteristics on the distribution of their decision-making power. 

Because the enterprise behavior originates from the executive decision, the executive characteristic is the key factor that affects the decision-making to cause the behavior, which produces the performance result. At the same time, the executive decision-making power decentralization, maybe makes managers not be arbitrary, and reduce negative events.

3. Providing convincing reasons for the controls such as CSR and other variables in the empirical model.

Response: The study mainly involves some controls variables： 

“As a result of the above theoretical research and influencing factors of EC and CPDI, the control variables are taken as the following factors including Enterprise’s Total Assets (that are owned or controlled for economic benefits), Employee Size (refers to the total number of enterprise jobs, including operators, managers, and employees), Current Assets (that can be liquidated within a year or a business cycle), Capital Reserve (refers to the provident fund formed by the acceptance of donations, the premium of equity and the appreciation of property), Operating Cost (refers to the cost of goods or services), and Asset-liability Ratio (reflecting the proportion of assets provided by the enterprise creditors). The data of these variables are obtained from the financial statements of listed companies to be taken as logarithms.”

4. Consider controlling some corporate governance variables such as the size of the board of directors, auditor quality, board independence, CEO characteristics, etc… (where the data is available)

Response: The variable of “the size of the board of directors”, has been calculated for the CPDI value, which divides the decision-making power of the company balance between the board and the manager in three steps. 

“In the first step, all executives are divided into three groups as director, supervisor, and management team to calculate TPF; Then, using 1, ..., n-1 to represent the seat number in an institution as the leader percentage of each group to calculate PPF; Finally, CPDI is equal to TPF×PPF. Based on previous research, the smaller entropy value represents the more dispersed power of decentralization and authorization for executors.”

5. Consider controlling for firm/industry fixed effects. The inclusion of time fixed effect is also encouraged.

Response: In the table of the paper report, Controls contains three types of control factors: 

(i) One is the control variable, which is respectively Enterprise’s Total Assets, Employee Size, Current Assets, Capital Reserve, Operating Cost, and Asset-liability Ratio; 

(ii) The second one is the firm fixed effect, which is to control the firm variable; 

(ii) The third one is the year fixed effect, that is, control of the time variable.

A sentence has been added at the bottom of Table 2 (The label in the paper): Controls include control variables, year fixed effect, and firm fixed effect.

At the same time, the complete regression results are shown in the table below, and we have omitted the control variables section of the paper.

Please see the appendix for the Table details, namely “Table 2 The effects of Entrepreneurial Characteristics on Corporate Social Responsibility”.

6. Using professional proofreading to improve comprehensive academic writing style and the quality of communication

Response: Following the reviewers’ advice, we carefully reviewed the manuscript, and the full text was re-combed, such as: 

“Panel A of Table 2 shows that the coefficient of Gender is positive and significant at 0.01, which is consistent with the conclusion that the increased proportion of female executives promotes CSR performance in literature. As the present proportion of female executives in listed companies is relatively small, they pay more attention to their social identity and personal accomplishment by CSR implementation. The mediator impact of CPDI has a significant positive correlation with female Gender characteristics (β=0.017, p<0.05, Model 2). It was indicated that female executives prefer stable structures, or they are not good at changing corporate governance. And it has a negative effect on CSR and is significant at 0.01 (Model 3). This support the Intrinsic Nature of CEOs towards CSR for Hypothesis H1. ”

For modifications to the full text, see the stamped attachment. 

Also, we changed the word type to Latex type as requested by the journal. We used Latex to organize all the contents of the paper, as shown in the appendix of the paper.

7. Minor errors need to be solved:

 In Hypothesis 1, there is a mistake after H1 "," (lines 102-103).

 In the conceptual model, there is an overlapping of the arrows (Figure 1)

 Table 1 can start at page 6, instead of the ending of the previous page.

Response: The Hypothesis and Figures have been modified. 

In the paper, the article segmentation, table layout, and graphic location have been following the requirements of the journal. we use Latex software to adjust the formats to match the journal requirements.

---

## [Decision Letter · Decision Letter 1]

11 Oct 2022

PONE-D-22-16972R1The decentralization effects of entrepreneurial characteristics on corporate social responsibilityPLOS ONE

Dear Dr. Hu,

Thank you for submitting your manuscript to PLOS ONE. After careful consideration, we feel that it has merit but does not fully meet PLOS ONE’s publication criteria as it currently stands. Therefore, we invite you to submit a revised version of the manuscript that addresses the points raised during the review process.

Thank you for incorporating the changes suggested by the reviewers. 

I suggest you to improve the literature review and conclusion part before the final acceptance of manuscript (minor revision).

We look forward to receiving your revised manuscript.

Kind regards,

Rana Muhammad Ammar Zahid, PhD

Academic Editor

PLOS ONE

Journal Requirements:

Additional Editor Comments 

Thank you for incorporating the changes suggested by the reviewers.

I suggest you to improve the literature review and conclusion part before the final acceptance of manuscript (minor revision).

1. The literature review part should incorporate recent research on the topic from the year 2021-2022, you may benefit from the following studies.

https://doi.org/10.3390/su131910662

https://doi.org/10.3389/fpsyg.2022.841163

https://doi.org/10.3389/fpsyg.2022.897444

2. In the conclusion part the implications looks very generic, make it specific to context based on the findings of your studies.

Reviewers' comments:

Reviewer's Responses to Questions

**Comments to the Author**

1. If the authors have adequately addressed your comments raised in a previous round of review and you feel that this manuscript is now acceptable for publication, you may indicate that here to bypass the “Comments to the Author” section, enter your conflict of interest statement in the “Confidential to Editor” section, and submit your "Accept" recommendation.

Reviewer #2: All comments have been addressed

2. Is the manuscript technically sound, and do the data support the conclusions?

Reviewer #2: Yes

3. Has the statistical analysis been performed appropriately and rigorously? 

Reviewer #2: Yes

4. Have the authors made all data underlying the findings in their manuscript fully available?

Reviewer #2: Yes

5. Is the manuscript presented in an intelligible fashion and written in standard English?

Reviewer #2: Yes

6. Review Comments to the Author

Reviewer #2: I believe that the authors improve in a greater extent the previos version as well as addressing the comments and suggestions of reviewers.

7. PLOS authors have the option to publish the peer review history of their article (what does this mean?). If published, this will include your full peer review and any attached files.

Reviewer #2: **Yes: **Jesus Barrena Martinez

---

## [Author Response · Author response to Decision Letter 1]

6 Nov 2022

1. The literature review part should incorporate recent research on the topic from the year 2021-2022, you may benefit from the following studies.

https://doi.org/10.3390/su131910662

https://doi.org/10.3389/fpsyg.2022.841163

https://doi.org/10.3389/fpsyg.2022.897444

Response: Thanks very much for the references provided by the reviewers. We downloaded the above three contacts and read them carefully. These papers have studied Corporate Social Responsibility Performance (CSRP) from different aspects, mainly as follows: 

(i) CEO characteristics positively moderate the relationship between corporate governance (CG) and social and environmental accountability (SEA).

(ii) Corporate Financial Performance (CFP) has a significant positive impact on CSRP, negatively moderated by CEO turnover and CEO duality. 

(iii) Tournament incentives motivate CEOs to be more socially responsible for CSRP, which will be positively affected through sub-national institutional contingencies. 

The above contents are of reference value to the contents we have studied. Therefore, the paper has been revised, and three articles are attached to the references. The specific modifications are as follows:

“The research results of Khan et al. [28] provide reliable evidence that tournament incentives motivate CEOs to be more socially responsible. ”

“The study by Zahid et al. [31] also found that CEO turnover and duality negatively moderate the relationship between corporate financial performance and CSR performance. Therefore, we must discuss the effectiveness of incentive features on CSR performance.”

“Khan et al. [36] found that the environmental performance of Chinese enterprises is affected by board size, independence, gender diversity, and CEO duality, which positively moderate the relationship between corporate governance and CSR. ” 

2. In the conclusion part the implications looks very generic, make it specific to context based on the findings of your studies.

Response: We have revised the conclusion as follows; see attachment of “Revised Manuscript with Track Changes”.

Conclusions and Policy Implementations

In this study, we used the personal data of CEOs and the financial data of listed companies to research the influence of Entrepreneurial Characteristics (EC) on Corporate Social Responsibility (CSR) with the moderating role of the Corporate Power Distribution Index (CPDI) based on the Upper Echelons Theory. In terms of emotion, morality, and value, we found that the Academy, Salary, and female Gender of EC have significant positive effects on CSR performance. Under the moderators of Age, Shares, and Academy characteristics, CPDI can significantly mediate the relationship between EC and CSR. Moreover, Politics background significantly mediates the impact of a Degree on CSR. The result provides that corporate strategy should optimize the power distribution of internal governance structure from entrepreneurs' self-interest to solve the adverse CSR events. As entrepreneurs often utilize their capacity and interests to carry out CSR activities, both the Incentive Feature and the Experience Characteristics can stimulate their responsibility to promote CSR performance. So there are two ways to improve CSR performance: 

On the one hand, it doesn't mean enterprises give up pursuing benefits to fulfill social responsibility activities. In contrast, they ought to optimize the corporate governance structure of the board of directors, supervisors, and managers to improve power distribution to avoid adverse CSR events even better for benefits. These checks and balances are affected by entrepreneurial characteristics, which adjust the corporate governance structure from different aspects of improving the educational learning degree of senior executives, introducing directors with academic or political backgrounds, and adequately allocating reasonable gender ratios. In addition, we should implement the incentive system with a management shareholding plan to reduce adverse CSR events; only when entrepreneurs know that the value is greater than the loss caused by CSR activities will it improve CSR performance correctly and efficiently. 

On the other hand, the governance of adverse CSR events should look into its root cause and symptoms of the problems for the regulatory authorities. In a short time, what plays an influential role is to give entrepreneurs material incentives, such as salary and equity, to encourage them to improve their CSR performance more actively and reward them for doing well. At the same time, regulators should note that CSR performance reflects more corporate ethics and entrepreneurship, which requires an arduous long-term project at the moral level. Therefore, the government ought to design a set of incentive and restraint mechanisms for entrepreneurs and establish a unique entrepreneurial credit evaluation system by combining morality, interests, and law to promote the enterprises to integrate CSR performance into corporate performance.

※ Please review your reference list to ensure that it is complete and correct. If you have cited papers that have been retracted, please include the rationale for doing so in the manuscript text, or remove these references and replace them with relevant current references. Any changes to the reference list should be mentioned in the rebuttal letter that accompanies your revised manuscript. If you need to cite a retracted article, indicate the article’s retracted status in the References list and also include a citation and full reference for the retraction notice.

Response: Thanks very much for your reminder. We proofread part of the paper's literature and references one by one. Then we replaced the problematic literature and added some valuable literature as follows.

1.Simon HA. Rational decision making in business organizations. The American Economic Review. 1979; 69: 493-513. http://www.jstor.org/stable/1808698

9.Kabir R, Thai HM. Does corporate governance shape the relationship between corporate social responsibility and financial performance? Pacific Accounting Review. 2017; 29: 227-258. https://doi.org/10.1108/PAR-10-2016-0091

10.Jackson SE, May KE, Whitney K. Understanding the dynamics of diversity in decision-making teams. Team Effectiveness and Decision Making in Organizations. 1995; 204: 203-261. https://www.researchgate.net/publication/275714101

12.Huang H, Zhao Z. The influence of political connection on corporate social responsibility: Evidence from listed private companies in China. International Journal of Corporate Social Responsibility. 2016; 1: 1-19. https://doi.org/10.1186/s40991-016-0007-3

14.Martínez JB, Fernández ML, Fernández PMR. Corporate social responsibility: Evolution through institutional and stakeholder perspectives. European journal of management and business economics. 2016; 25: 8-14. https://doi.org/10.1016/j.redee.2015.11.002

22.Beji R, Yousfi O, Loukil N, Omri A. Board diversity and corporate social responsibility: Empirical evidence from France. Journal of Business Ethics, 2021; 173: 133-155. https://doi.org/10.1007/s10551-020-04522-4

26.Martínez JB, Fernández ML, Moreno CM, Fernández PMR. Corporate social responsibility in the process of attracting college graduates. Corporate Social Responsibility and Environmental Management, 2015; 22: 408-423. https://doi.org/10.1002/csr.1355

28.Khan M K, Ali S, Zahid R M A, et al. Does whipping tournament incentives spur CSR performance? An empirical evidence from Chinese Sub-national Institutional Contingencies. Frontiers in psychology. 2022; 13: 841163. https://doi.org/10.3389/fpsyg.2022.841163

30.Liu J, Zhu Y. Executive shareholding, internal control and inefficient input of corporate social responsibility. Journal of Shandong University of Finance and Economics. 2018; 30: 55-65. https://doi.org/10.3969/j.issn.1008-2670.2018.06.007

31.Zahid RMA, Khurshid M, Khan W. Do chief executives matter in corporate financial and social responsibility performance nexus? A dynamic model analysis of Chinese firms. Frontiers in Psychology, 2022; 13. https://doi.org/10.3389/fpsyg.2022.897444

33.Ahmed AS, Duellman S. Managerial overconfidence and accounting conservatism. Journal of Accounting Research. 2013; 51: 1-30. https://doi.org/10.1111/j.1475-679X.2012.00467.x

37.Khan MK, Zahid RMA, Saleem A, Sági J. Board composition and social & environmental accountability: A dynamic model analysis of Chinese firms. Sustainability, 2021; 13: 10662. https://doi.org/10.3390/su131910662

※ While revising your submission, please upload your figure files to the Preflight Analysis and Conversion Engine (PACE) digital diagnostic tool, https://pacev2.apexcovantage.com/. PACE helps ensure that figures meet PLOS requirements. 

Response: The four pictures in the article have been uploaded to PACE and passed the inspection. And they are packaged into the fig.tif file.

Fig 1.

(a)Shares (b)Age (c)Academy

Fig 2.

---

## [Editor Report · Decision Letter 2]

9 Nov 2022

The decentralization effects of entrepreneurial characteristics on corporate social responsibility

PONE-D-22-16972R2

Dear Dr. Hu,

We’re pleased to inform you that your manuscript has been judged scientifically suitable for publication and will be formally accepted for publication once it meets all outstanding technical requirements.

Kind regards,

Rana Muhammad Ammar Zahid, PhD

Academic Editor

PLOS ONE

Additional Editor Comments (optional):

Thank you for incorporating suggested changes.
---

## [Editor Report · Acceptance letter]

16 Nov 2022

PONE-D-22-16972R2 

The decentralization effects of entrepreneurial characteristics on corporate social responsibility 

Dear Dr. Hu:

I'm pleased to inform you that your manuscript has been deemed suitable for publication in PLOS ONE. Congratulations! Your manuscript is now with our production department. 

Kind regards, 

on behalf of

Dr. Rana Muhammad Ammar Zahid 

Academic Editor

PLOS ONE